# Unveiling a Surgical Revolution: The Use of Conventional Histology versus Ex Vivo Fusion Confocal Microscopy in Breast Cancer Surgery

**DOI:** 10.3390/cells13201692

**Published:** 2024-10-12

**Authors:** Daniel Humaran, Javiera Pérez-Anker, Pedro L. Fernández, Lidia Blay, Iciar Pascual, Eva Castellà, Laia Pérez, Susana Puig, Josep Malvehy, Joan F. Julián

**Affiliations:** 1Department of General and Digestive Surgery, Hospital Universitari Germans Trias I Pujol, Universitat Autònoma de Barcelona (UAB), 08916 Badalona, Spain; lydia.blay@gmail.com (L.B.); irizarp@gmail.com (I.P.); jfjulian.germanstrias@gencat.cat (J.F.J.); 2Department of Surgery, Universitat Autònoma de Barcelona (UAB), 08035 Barcelona, Spain; 3Department of Dermatology, Hospital Clínic de Barcelona, Universitat de Barcelona (UB), 08036 Barcelona, Spain; javiperezanker@gmail.com (J.P.-A.); susipuig@gmail.com (S.P.); jmalvehy@gmail.com (J.M.); 4Department of Pathology, Institut de Recerca Germans Trias I Pujol, Hospital Universitari Germans Trias I Pujol, Universitat Autònoma de Barcelona (UAB), 08916 Badalona, Spain; plfernandez.germanstrias@gencat.cat (P.L.F.); ecastella.germanstrias@gencat.cat (E.C.); laiaproca@gmail.com (L.P.)

**Keywords:** breast cancer detection, breast conserving surgery, ex vivo fusion confocal microscopy, surgical pathology

## Abstract

Ex vivo fusion confocal microscopy (EVFCM) enables the rapid examination of breast tissue and has the potential to reduce the surgical margins and the necessity for further surgeries. Traditional methods, such as frozen section analysis, are limited by the distortion of tissue and artefacts, leading to false negatives and the need for additional surgeries. This study on observational diagnostic accuracy evaluated the ability of EVFCM to detect breast cancer. A total of 36 breast tissue samples, comprising 20 non-neoplastic and 16 neoplastic cases, were analysed using EVFCM and compared to the results obtained from routine histopathology. A Mohs surgeon experienced in EVFCM (evaluator A) and two breast pathologists unfamiliar with EVFCM (evaluators B and C) performed blinded analyses. EVFCM showed high concordance with the histopathology and the detection of neoplasia, with significant kappa values (*p* < 0.001). Evaluator A achieved 100% sensitivity and specificity. Evaluators B and C achieved a sensitivity of >87%, a specificity of >94%, positive predictive values of >95%, and negative predictive values of 81% and 94%, respectively. EVFCM therefore offers a promising technique for the assessment of margins in breast-conserving surgery. Its widespread adoption could significantly reduce re-excisions, lower healthcare costs, and improve cosmetic and oncological outcomes.

## 1. Introduction

Globally, breast cancer is the most frequently diagnosed cancer in women, with over 2.29 million new cases reported worldwide in 2022 [1]. Consequently, a significant number of women undergo breast surgery each year. While mastectomy used to be the predominant surgical treatment, breast-conserving surgery now tends to be performed whenever feasible [2].

Breast surgeons are often confronted with a dilemma, either generating a minimal breast defect during the removal of tumours or performing a larger resection, which can negatively affect cosmetic outcomes [3]. Intraoperatively, the need for a rapid and reliable histological assessment of breast cancer is crucial as it is imperative to achieve negative margins in breast-conserving surgery [4], significantly impacting the success of the procedure. To ensure a thorough oncological resection, the margins of the tumour excision are assessed via postoperative histopathological examination. If cancer cells are detected at the margins of the excised tissue, it is typically advised that a re-excision of the surgical bed, a new tumorectomy, or even a mastectomy [5] be performed due to the substantial risk of tumour recurrence.

A significant proportion of breast cancer patients undergoing breast-conserving surgery require additional procedures due to positive margins, with at least 20% needing more than one surgery as part of their treatment plan [6,7,8]. This situation results in higher healthcare costs, worsened oncological outcomes [9], increased morbidity, compromised cosmetic outcomes, and a negative impact on the psychological well-being of the women involved [10].

The methods that enable the immediate assessment of tissue are highly valued in procedures that demand prompt evaluation, such as intraoperative margin evaluation [10], during the surgical treatment of breast cancer. The use of intraoperative frozen section analysis is considered as the gold standard microscopic technique for the rapid evaluation of histologic margins, particularly in oncologic procedures [11,12]. However, artefacts frequently occur in frozen sections, and these can distort the tissue structure and compromise the optimal preservation of the specimen. Furthermore, frozen sections depend on specific laboratory resources and are difficult to evaluate [11], particularly those composed predominantly of adipose tissue such as breast samples [13,14,15]. Additionally, frozen section analysis has demonstrated limitations in terms of its sampling variability and morphological and time constraints, which can result in false negatives [16,17].

Ex vivo fusion confocal microscopy (EVFCM) is a promising technique that provides several benefits compared to frozen section analysis, addressing the weaknesses of conventional histologic methods. It provides the real-time imaging of tissue structure at a cellular resolution, eliminating the need for complex tissue preparation [18,19,20,21]. This technology is particularly valuable for tissues with a high concentration of adipose cells, such as breast samples [11,22], in the assessment of intraoperative margins, in interventional radiology procedures [13], and even in the evaluation of core needle biopsies [23]. Additionally, it is worth noting that the confocal microscopy method requires minor sample processing and does not lead to any tissue loss, in contrast to frozen sectioning [13]. EVFCM also provides images of the structural characteristics of breast samples, similar to the standard histopathologic analysis of haematoxylin–eosin (H&E)-stained tissue slides [23,24]. Moreover, EVFCM features a fluorescence imaging mode, enabling the rapid identification of regions in the sample that potentially present positive margins during intraoperative surgical resections [23,24,25]. This preserves the integrity of tissues without compromising routine histologic evaluation [20,22].

In this study, we conducted an observational diagnostic accuracy evaluation of EVFCM with regard to the diagnosis of breast cancer. The main aim of this study was to measure the accuracy of EVFCM in distinguishing between neoplastic and non-neoplastic breast tissue, comparing its results to the gold standard of histopathological analysis (H&E staining).

The first objective of this study was to evaluate the ability of expert breast cancer histopathologists with no prior experience of EVFCM to identify whether neoplasia was present or absent using only EVFCM images. The second objective was to assess their ability to accurately identify the histological subtypes. Finally, the third objective was to evaluate the diagnostic capabilities of an EVFCM researcher with no previous experience in breast pathologies.

This study also aimed to contribute to the advancement of breast cancer diagnostics by validating EVFCM as a minimally invasive tool for the real-time assessment of margins, potentially reducing re-excision rates and improving patient outcomes.

## 2. Materials and Methods

### 2.1. Tissue Selection, Staining, and Imaging

A total of 36 frozen breast tissue samples, comprising 20 non-neoplastic and 16 neoplastic specimens, were acquired from the biobank of our institution once this study had been evaluated by the review board and informed consent had been obtained. This study adhered to the guidelines set by the Ethics Committee.

The neoplastic cases predominantly consisted of invasive ductal carcinomas (*n* = 12) (Figure 1, Figure 2 and Figure 3), but also included one case each of phyllodes tumour, ductal carcinoma in situ, invasive lobular carcinoma, and mucinous carcinoma (Figure 4).

Each sample was completely immersed in saline solution and stained in 50% acetic acid (Panreac Química, Barcelona, Spain) for 20 s; this was followed by a wash in saline solution and additional staining in 1 mmol/L acridine orange (Sigma-Aldrich, Merck KGaA, Madrid, Spain) for 20 s. Acridine orange is a dye that binds to nucleic acids and emits green fluorescence when attached to DNA and RNA, with a peak fluorescence emission when excited at 488 nm [26,27]. Acetic acid was used to enhance nuclear detail by inducing the compaction of chromatin, which improves reflectance signals and visualisation. This dye combination increases the visibility of both stromal and tumour structures, enhancing the contrast and definition of the tissue architecture in EVFCM without damaging the samples, allowing for subsequent histopathological analysis [28].

For imaging, the samples were scanned using the VivaScope^®^2500M-G4 device [29]. This device incorporates dual lasers that operate at distinct wavelengths: fluorescence (488 nm) and reflectance (638 nm). Scanning was performed using the fusion mode, which merges the fluorescence and reflectance channels to generate a comprehensive, high-resolution image of the tissue architecture and cellular morphology. This mode is especially useful for distinguishing between neoplastic and non-neoplastic tissue structures, enhancing the information provided by the acetic acid and acridine orange stains.

After scanning, the samples underwent routine histopathological processing, including fixation in formalin, embedment in paraffin, tissue sectioning, and staining with H&E for definitive diagnosis.

### 2.2. Confocal Image Evaluation and Pathologist Assessments

Three evaluators independently assessed the EVFCM images: a Mohs surgeon dermatologist experienced in EVFCM, with no prior experience in breast pathology (evaluator A), and two expert breast pathologists with no prior experience in EVFCM (evaluators B and C). The evaluators were blinded to the final histopathological diagnoses, and they classified the images as either neoplastic or non-neoplastic. The evaluators were instructed to rely solely on the EVFCM images for their assessments, and no list of potential histological subtypes was provided; this was to avoid bias in the interpretation.

Each sample was evaluated twice by each pathologist, with a 12-month washout period between the two evaluations to prevent recall bias. Prior to the second evaluation, one of the pathologists (evaluator B) underwent a brief, 4 h training session on the appearance and morphology of cutaneous carcinomas in EVFCM. In contrast, evaluator C did not receive additional training, but was allowed to examine all the histopathology samples without their corresponding confocal images. The pathologists were asked about the presence or absence of neoplasia during both evaluations and, if observed, the subtype of the visualised neoplasia was noted.

The evaluators were not shown the histopathology results until all evaluations were completed. After the second confocal evaluation, the pathologists reviewed the permanent H&E slides, and the final consensus diagnosis was determined based on the H&E findings. Each EVFCM image was compared to the corresponding histopathological section to confirm the accuracy.

### 2.3. Statistical Analysis

An exploratory data analysis was conducted using frequency and percentage distributions. The sensitivity, specificity, positive predictive value (PPV), and negative predictive value (NPV) were calculated [30]. To assess the correlation and agreement between the evaluators, the kappa index of concordance was computed [31]. The final evaluation of the H&E-stained slides produced by the two expert pathologists served as the gold standard. A significance level of 5% was adopted. The PASW statistical software, Version 18.0. (SPSS Inc., Chicago, IL, USA) was used for data analysis.

## 3. Results

### 3.1. Neoplasia Presence Accuracy

The investigators’ ability to identify the presence or absence of neoplasia in EVCFM images is summarised in Table 1. There was a high level of agreement and concordance in all comparisons. All κ values were statistically significant (*p* < 0.001). Investigator A detected neoplasia with a 100% sensitivity and specificity. Regarding the ability of the pathologists without any previous knowledge of EVFCM (B and C) to diagnose the presence or absence of neoplasia, in the first evaluation, their sensitivity was higher than 87% (87% and 95%), and their specificity was higher than 94% (100% and 94%), with κ values of 0.828 and 0.885, respectively. Both PPVs were higher than 95% (100% and 95%), and the NPVs were 81% and 94%, respectively. After receiving the short training course on EVFCM, in the second evaluation, the sensitivity of evaluator B increased by 3% (from 87% to 90%), and their NPV increased by 7% (from 81% to 88%), with a specificity of 93% and a κ value of 0.830. The lowest value observed for the presence of neoplasia was 85.2% when the second assessment of evaluator C was compared with the histopathology.

### 3.2. Histologic Subtype Recognition

The results of the histopathologic subtype recognition using EVFCM images are summarised in Table 2. Considering that the most predominant subtype in the sample was the invasive ductal carcinoma, evaluator B achieved an accuracy of 86.1% in the first assessment, with a sensitivity of 82% and a specificity of 88%. The PPV was 75%, the NPV was 92%, and the κ value was 0.681. In the second evaluation, the accuracy increased by 2.8%, with an improvement in sensitivity of 1% (from 82% to 83%) and an improvement in specificity of 4% (from 88% to 92%). The PPV also increased by 8% (from 75% to 83%), while the NPV remained at 92%. The κ value improved from 0.681 to 0.750. Otherwise, in the first evaluation, evaluator C showed excellent results, achieving an accuracy of 97.2%, with a κ value of 0.936. The sensitivity reached 100%, while the specificity was 96%, with a PPV of 92% and an NPV of 100%.

## 4. Discussion

Based on the numerous advantages of confocal microscopy, we conducted an initial study utilising EVFCM in breast cancer. The main goals of this research were to assess the ability of EVFCM to identify neoplasia and determine the histologic subtypes with diagnostic accuracy. To achieve this, we correlated our EVFCM images with the corresponding permanent histological sections, enabling a direct morphological comparison.

Our results reveal a high level of agreement and concordance for the diagnosis of breast cancer, as demonstrated by an excellent kappa value and a statistically significant *p*-value of <0.001 for all comparisons of breast samples. These findings indicate that an evaluator with prior knowledge of EVFCM, despite lacking specific training in breast tissue analysis, can detect the presence or absence of neoplasia with excellent diagnostic accuracy, with a reported accuracy of 100%. Furthermore, expert pathologists specialising in breast cancer, even without prior experience in EVFCM, exhibited high sensitivity and specificity rates (greater than 87% and 94%, respectively) during their initial evaluations. Notably, both pathologists demonstrated the ability to recognise invasive ductal carcinoma and other subtypes with impressive precision.

Our study also observed that a few hours of training can significantly improve individuals’ ability to diagnose neoplastic samples. For instance, the sensitivity of evaluator B increased by 3% (from 87% to 90%), and their NPV increased by 7% (from 81% to 88%) after training. Post-training, the ability of evaluator B to identify histological subtypes with accuracy increased by 2.8% (from 86.1% to 88.9%), their sensitivity increased by 1% (from 82% to 83%), their specificity increased by 4% (from 88% to 92%), and their PPV increased by 8% (from 75% to 83%).

In line with the findings of Panarello [32], Bertoni [33], and Shavlokhova et al. [34], our results suggest that the learning curve for interpreting confocal images is relatively short. Prior knowledge of EVFCM can enhance individuals’ ability to accurately identify neoplasia, although distinguishing specific subtypes can remain challenging for those with limited experience of EVFCM.

Bertoni et al. [33] found similar trends when using EVFCM to evaluate prostate biopsies, where the agreement between EVFCM and the histopathological diagnosis improved significantly after a brief training period. Their kappa values increased from 0.68 and 0.79 in the first evaluation to 0.87 in the second, and the ROC curve improved from 0.87 to 0.93; as such, the agreement rates increased from 86% to 95%. Similarly, Shavlokhova et al. [34] reported that there is a short learning curve for confocal image interpretation in oral tissues, with the agreement increasing from 89% in the first evaluation to 97% in the second; this was accompanied by significant improvements in sensitivity and specificity. In 2023, the HIBISCUSS project [7] demonstrated that skill in breast cancer detection could be rapidly acquired, with a pathologist achieving an accuracy of 99.6% following structured training. Surgeons, after a brief training programme, showed a notable improvement, with their sensitivity increasing from 83% to 98% and specificity rising from 84% to 87%. These trends underscore the potential for skill in the interpretation of confocal images to be rapidly acquired, and further highlight the importance of structured training in enhancing diagnostic accuracy.

Conversely, evaluator C, who did not receive training, showed no improvement in the second evaluation. While confusion stemming from exposure to histopathology samples without their corresponding confocal images may have contributed to these inferior results, it is important to recognise that the lack of structured training likely hindered evaluator C’s ability to accurately interpret the confocal images. Additionally, the 12-month washout period before the second evaluation likely disrupted the continuity of the learning curve for evaluator C. This situation illustrates the critical role of structured training programmes in mastering EVFCM.

Neoplastic tissues were readily distinguishable from normal structures using EVFCM, which offers an enhanced fluorescence contrast and improved image quality without compromising the subsequent histological evaluation. Our results demonstrated that using EVFCM to evaluate breast tissue provides high accuracy compared to conventional histological approaches, without the occurrence of artefacts associated with frozen sections; therefore, excellent rates of concordance, sensitivity and specificity were achieved, along with high kappa values, PPVs, and NPVs.

We utilised the VivaScope^®^2500M-G4 device [29], which features dual scanners for fluorescence and reflectance that operate simultaneously. This device offers a significantly faster and more reliable option for histological studies compared to previously available devices. Furthermore, the combination of acridine orange and acetic acid staining proved highly valuable, enabling the visualisation of both cytoplasmatic and nuclear details, as demonstrated in previous studies [11,12,13,23,25]. When compared with frozen sections, the EVFCM images provided superior tissue integrity, excellent morphological detail, and a satisfactory staining quality, not just in adipose tissue, but also in tumour regions. These findings suggest that EVFCM could potentially compete with frozen section analysis, particularly in breast tissue.

Recent studies further support the potential use of confocal microscopy in breast cancer diagnosis, with encouraging outcomes being achieved.

Brachtel et al. [6] reported that intraoperative margin assessment has a high diagnostic accuracy, with a sensitivity of 0.91, a specificity of 0.93, a PPV of 0.95, an NPV of 0.87 and strong intra- and interobserver agreement (0.87 and 0.84, respectively). These results align closely with the findings of this study. Krishnamurthy et al. [13] demonstrated the feasibility of using confocal microscopy in breast surgical resections, finding that confocal images closely resemble conventional histopathology, while offering more rapid acquisition and high-resolution images. Their use of acridine orange for enhanced tissue recognition also mirrors our staining approach using EVFCM, further supporting its potential use in breast cancer diagnosis. Notably, their greyscale confocal images differed from our pseudo-H&E images, which offer improved neoplastic tissue definition. Scimone et al. [24] reported a perfect positive predictive value (PPV = 1) and a strong NPV of 0.83, detecting positive margins; this underscores the accuracy of intraoperative evaluation, with results similar to the high concordance and diagnostic performance seen in our study. Nackenhorst et al. [11] also confirmed the ability to detect breast mastectomy samples with high diagnostic accuracy; this highlights its ease of use and reliability, with no incorrect assessment of tumour invasion. The HIBISCUSS project emphasizes the clinical utility of EVFCM for both pathologists and surgeons in breast cancer surgery [7]. These studies, along with our findings, suggest that EVFCM has the potential to improve the accuracy and efficiency of breast cancer surgery.

EVFCM enhances both surgical precision and diagnostic accuracy, while virtual platforms, such as those proposed by Żydowicz et al. [35] could complement these advancements by optimising patient care. Integrating artificial intelligence (AI) and deep learning systems could further enhance EVFCM’s capabilities by automating the identification of neoplastic cells, thus improving diagnostic accuracy and reducing the workload of the pathologist. This application of AI, particularly in intraoperative margin assessments, could decrease the rate of positive margins and re-excisions, maximising the benefits of EVFCM in surgery. The integration of AI, EVFCM, and virtual platforms offers a pathway for transforming breast cancer care into a more precise, efficient, and patient-centred model.

Based on all these data and our results, we advocate the use of EVFCM as a highly effective method for intraoperative margin assessment in breast-conserving surgery due to its high accuracy in diagnosis and applicability. Therefore, rapid intra-operative margin analysis using EVFCM could help decrease the number of patients undergoing mastectomies or requiring additional resections, increasing the rates of breast-conserving and expanding the surgical margins if necessary. However, the lack of standardisation currently hinders the widespread adoption of this procedure as a standard of care in many breast cancer centres in Europe. Despite these promising results, further studies with a larger number of samples that represent various histological types are necessary to analyse the sensitivity and specificity of EVFCM in recognising different subtypes of tumours; these results should then be compared with our initial data. One of the limitations of this report was the limited number of neoplastic and non-neoplastic samples used. Nonetheless, we are confident that EVFCM could play a pivotal role in advancing breast cancer surgery, and we are committed to further research that establishes it as a standard procedure in breast-conserving oncologic surgeries.

## 5. Conclusions

In conclusion, we have presented our experience and results comparing images obtained through EVFCM, a method involving minimal tissue processing, for the identification of various neoplastic and non-neoplastic breast lesions. EVFCM offers a cost-effective approach to further evaluation, as well as optimal tissue preservation. The high-definition images acquired through EVFCM enable the detailed assessment of nuclear characteristics and cellular morphology, which correspond closely to those seen in histologic images stained with H&E. This study, based on a visual assessment, demonstrated that there is a high level of agreement and correlation between EVFCM and conventional histology in the differentiation of neoplastic from non-neoplastic lesions.

EVFCM is able to identify breast neoplasms with high diagnostic accuracy and effectively identify histological subtypes, achieving high concordance, sensitivity and specificity rates. Importantly, training in EVFCM enables evaluators to achieve a diagnostic capacity of 100% for breast tissues, even without prior experience in breast pathology. Based on our findings, we propose that EVFCM could serve as a valuable method for the evaluation of intraoperative margins, aiding breast-conserving surgery and potentially reducing the need for reoperations, mastectomies, or additional resections; thus, breast conservation can be promoted.

The widespread adoption of EVFCM could significantly improve healthcare outcomes by reducing re-excisions, which would lower healthcare costs and improve cosmetic results for patients. By minimising the need for additional surgeries, EVFCM could also enhance recovery and the psychological well-being of patients, as fewer surgeries would reduce the physical and emotional toll on patients. These combined benefits highlight the potential of EVFCM to transform breast cancer surgery, offering clinical, economic, and psychological advantages that enhance the overall quality of patient care. We believe that EVFCM could play a pivotal role in revolutionising the paradigm of breast cancer surgery.

## Figures and Tables

**Figure 1 cells-13-01692-f001:**
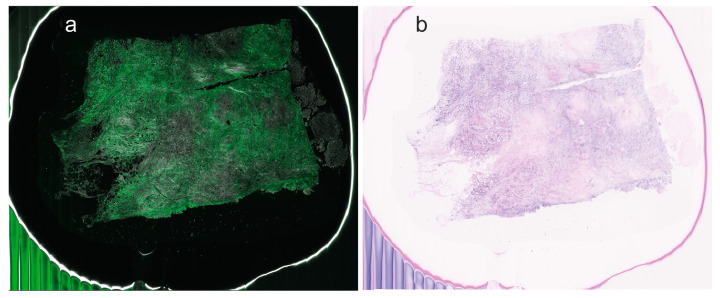
Invasive ductal carcinoma scanned using ex vivo fusion confocal microscopy VivaScope^®^2500M-G4 (Mavig GmbH, Munich, Germany; Caliber I.D., Rochester, NY, USA). (**a**) Combined fluorescence and reflectance signals. (**b**) Conversion of fusion signalling to pseudo-coloured haematoxylin–eosin.

**Figure 2 cells-13-01692-f002:**
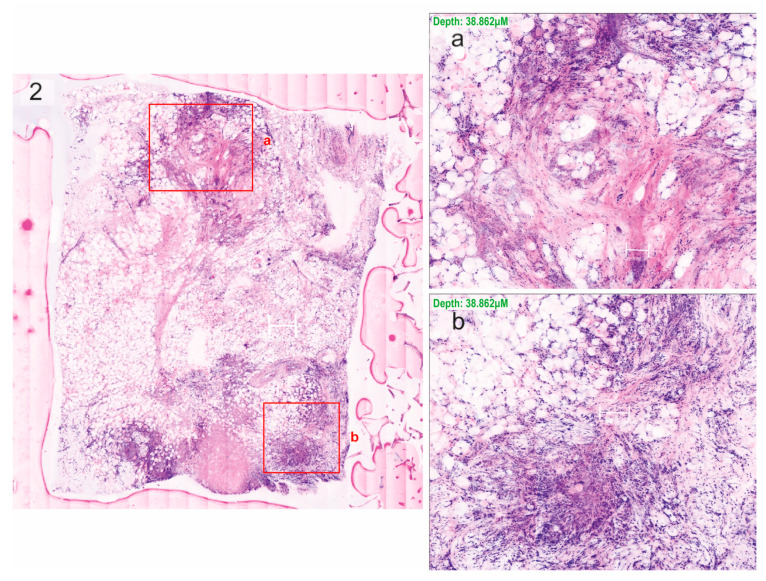
Invasive ductal carcinoma scanned using ex vivo fusion confocal microscopy VivaScope^®^2500M-G4, pseudo-coloured to resemble haematoxylin–eosin staining, with regions of interest (boxes). (**a**,**b**) Zoom in on those regions for the identification of cancer cells.

**Figure 3 cells-13-01692-f003:**
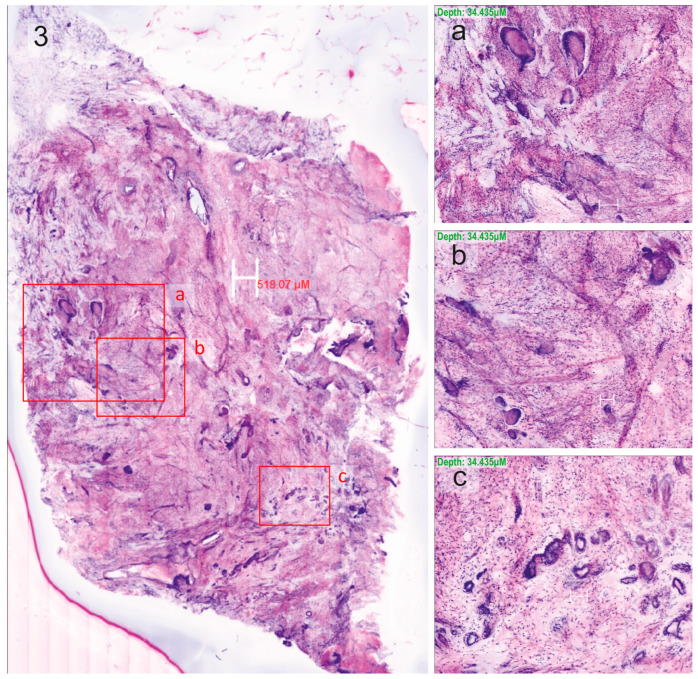
Invasive ductal carcinoma scanned using ex vivo fusion confocal microscopy VivaScope^®^2500M-G4, pseudo-coloured to resemble haematoxylin–eosin staining, with regions of interest (boxes). (**a**–**c**) Zoom in on those regions for the identification of cancer cells.

**Figure 4 cells-13-01692-f004:**
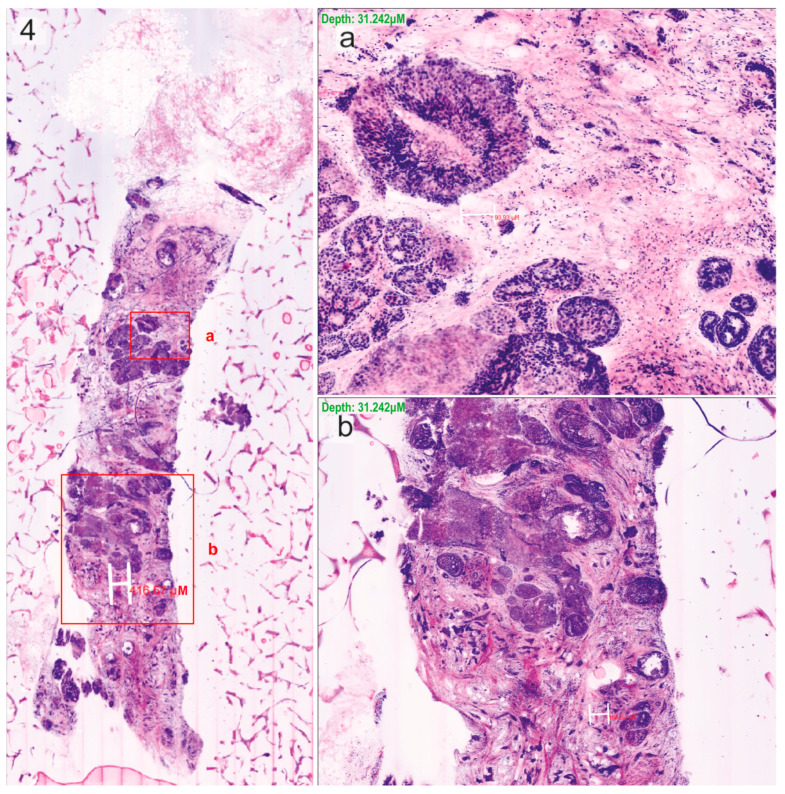
Mucinous carcinoma scanned using ex vivo fusion confocal microscopy VivaScope^®^2500M-G4, pseudo-coloured to resemble haematoxylin–eosin, with regions of interest (boxes). (**a**,**b**) Zoom in on those regions for the identification of cancer cells.

**Table 1 cells-13-01692-t001:** Neoplasia presence: concordance between the evaluators.

Evaluators	Samples	Agreement (%)	Kappa (K)	*p*-Value of K	Sensitivity (%)	Specificity (%)	PPV (%)	NPV (%)
A vs. HP	36	100	1	<0.001	100	100	100	100
B 1st vs. HP	36	91.7	0.828	<0.001	87	100	100	81
B 2nd vs. HP	36	91.7	0.830	<0.001	90	93	95	88
C 1st vs. HP	36	94.3	0.885	<0.001	95	94	95	94
C 2nd vs. HP	36	85.2	0.715	<0.001	94	78	80	93

Sensitivity and specificity of each investigator and kappa value. HP: histopathology. Evaluators: A (dermatologist experienced in EVCFM), B (pathologist trained for a second evaluation), C (pathologist without training for a second evaluation). 1st (first evaluation); 2nd (second evaluation). PPV: positive predictive value. NPV: negative predictive value.

**Table 2 cells-13-01692-t002:** Histologic Subtype Recognition: Concordance between Investigators.

Evaluators	Samples	Agreement (%)	Kappa (K)	Sensitivity (%)	Specificity (%)	PPV (%)	NPV (%)
B 1st vs. HP	36	86.1	0.681	82	88	75	92
B 2nd vs. HP	36	88.9	0.750	83	92	83	92
C 1st vs. HP	36	97.2	0.936	100	96	92	100
C 2nd vs. HP	36	83.3	0.640	71	91	83	83

Sensitivity and specificity of each investigator and kappa values. HP: histopathology. Evaluators: B (pathologist trained for a second evaluation), C (pathologist without training for a second evaluation). 1st (first evaluation), 2nd (second evaluation). PPV: positive predictive value. NPV: negative predictive value.

## Data Availability

The data presented in this study that support the findings of this study are available upon request. Due to privacy and ethical considerations, access to the data will be granted after a formal request to the corresponding author and upon completion of any necessary data sharing agreements.

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
