# Peer review of "Unveiling a Surgical Revolution: The Use of Conventional Histology versus Ex Vivo Fusion Confocal Microscopy in Breast Cancer Surgery"

_cells, 2024, doi:10.3390/cells13201692_

Round 1
Reviewer 1 Report
Comments and Suggestions for Authors
The paper titled "Unveiling a surgical revolution: single-blind assessment of conventional histology and ex-vivo fusion confocal microscopy in breast cancer surgery" presented a assessment technique for breast cancer diagnosis. The paper needs major revision in order to get published in Cells.
Abstract requires the limitations of the existing techniques and why we need the proposed assessment technique. It should present the future directions of the proposed method.
The paper requires in-depth proofread. I have noticed many flaws in sentence formations.
In introduction, there is no study contributions. A clearly defined objectives or contributions should be listed at the end of the introduction.
In section 2, only figures are displayed, there is no in-depth information regarding the proposed methodology. This section requires a detailed methodology to explain the study approach.
How specificity, sensitivity and Kappa values are computed?
References 6-12 are too old, the authors should consider to replace the recent references.
Comments on the Quality of English LanguageExtensive English editing is required.
Author Response
- Comment 1. The paper titled "Unveiling a surgical revolution: single-blind assessment of conventional histology and ex-vivo fusion confocal microscopy in breast cancer surgery" presented an assessment technique for breast cancer diagnosis. The paper needs major revision in order to get published in Cells.
Response 1:
Dear reviewer,
We would like to sincerely thank you for your valuable comments on our manuscript. We have carefully reviewed your feedback and have worked diligently to address each of the points raised.
We are pleased to inform you that we have implemented the necessary revisions to enhance the quality of the manuscript in accordance with your recommendations. We believe these revisions have significantly strengthened the manuscript, and we are confident that it now aligns with the high standards expected by Cells.
We greatly appreciate the time and effort you have dedicated to evaluating our work. Your feedback has been instrumental in improving the clarity and scientific rigor of the paper. We trust that the revised version will meet your expectations.
Below, we provide a detailed breakdown of each of your comments along with the respective changes made in response.
- Comment 2. Abstract requires the limitations of the existing techniques and why we need the proposed assessment technique. It should present the future directions of the proposed method.
Response 2:
Thank you for your insightful feedback regarding the abstract. We appreciate your suggestion to include more detail on the limitations of existing techniques, the justification for the proposed method (EVFCM), and the future directions of this approach. In response, we have made the following revisions to the abstract:
Limitations of Existing Techniques: “Traditional methods, such as frozen section analysis, are limited by the distortion of tissue and artefacts, leading to false negatives and the need for additional surgeries.” This establishes a clear rationale for exploring alternative techniques such as EVFCM, which addresses these shortcomings.
Why EVFCM is Needed: To address this issue we explained: “Ex vivo fusion confocal microscopy (EVFCM) enables the rapid examination of breast tissue and has the potential to reduce the surgical margins and the necessity for further surgeries” and “EVFCM showed high concordance with the histopathology and the detection of neoplasia, with significant kappa values (p<0.001).”
Future Directions: In response to your request for a discussion of the future directions of EVFCM, we included a section on the abstract: “ EVFCM therefore offers a promising technique for the assessment of margins in breast-conserving surgery. Its widespread adoption could significantly reduce re-excisions, lower healthcare costs, and improve cosmetic and oncological outcomes.“ We trust this revision address the importance of EVFCM in the current surgical landscape and offer a clear vision of its potential future impact in breast cancer surgery.
We believe these revisions have significantly improved the clarity and relevance of the abstract, and we hope that it now better aligns with your expectations.
The revised abstract is as follows:
“ Ex vivo fusion confocal microscopy (EVFCM) enables the rapid examination of breast tissue and has the potential to reduce the surgical margins and the necessity for further surgeries. Traditional methods, such as frozen section analysis, are limited by the distortion of tissue and artefacts, leading to false negatives and the need for additional surgeries. This study on observational diagnostic accuracy evaluated the ability of EVFCM to detect breast cancer. A total of 36 breast tissue samples, comprising 20 non-neoplastic and 16 neoplastic cases, were analysed using EVFCM and compared to the results obtained from routine histopathology. A Mohs surgeon experienced in EVFCM (evaluator A) and two breast pathologists unfamiliar with EVFCM (evaluators B and C) performed blinded analyses. EVFCM showed high concordance with the histopathology and the detection of neoplasia, with significant kappa values (p<0.001). Evaluator A achieved 100% sensitivity and specificity. Evaluators B and C achieved a sensitivity of >87%, a specificity of >94%, positive predictive values of >95%, and negative predictive values of 81% and 94%, respectively. EVFCM therefore offers a promising technique for the assessment of margins in breast-conserving surgery. Its widespread adoption could significantly reduce re-excisions, lower healthcare costs, and improve cosmetic and oncological outcomes. “
- Comment 3. The paper requires in-depth proofread. I have noticed many flaws in sentence formations.
Response 3:
Thank you for your valuable feedback regarding the language and sentence formation in our manuscript. We have carefully considered your comments and conducted a thorough proofreading of the entire manuscript to ensure clarity and fluency. Additionally, in response to your suggestion, we utilised the English Editing service provided by MDPI to address any issues in sentence construction and overall readability. We are confident that these revisions have significantly improved the manuscript and believe it now meets the required English language standards.
- Comment 4. In introduction, there is no study contributions. A clearly defined objectives or contributions should be listed at the end of the introduction.
Response 4:
Thank you for your valuable feedback regarding the need for clearly defined objectives and contributions in the introduction. We have carefully revised the introduction to ensure that the study’s goals and contributions are explicitly stated.
At the end of the introduction, we have now included the following:
“In this study, we conducted an observational diagnostic accuracy evaluation of EVFCM with regard to the diagnosis of breast cancer. The main aim of this study was to measure the accuracy of EVFCM in distinguishing between neoplastic and non-neoplastic breast tissue, comparing its results to the gold standard of histopathological analysis (H&E staining).
The first objective of this study was to evaluate the ability of expert breast cancer histopathologists with no prior experience of EVFCM to identify whether neoplasia was present or absent using only EVFCM images. The second objective was to assess their ability to accurately identify the histological subtypes. Finally, the third objective was to evaluate the diagnostic capabilities of an EVFCM researcher with no previous experience in breast pathologies.
This study also aimed to contribute to the advancement of breast cancer diagnostics by validating EVFCM as a minimally invasive tool for the real-time assessment of margins, potentially reducing re-excision rates and improving patient outcomes.”
We believe this revision clearly addresses the objectives and contributions of the study, as per your suggestion, and helps clarify the purpose and significance of the research.
- Comment 5. In section 2, only figures are displayed, there is no in-depth information regarding the proposed methodology. This section requires a detailed methodology to explain the study approach.
Response 5:
Thank you for your valuable feedback regarding the need for a more detailed explanation of the methodology in Section 2. We have carefully revised the Materials and Methods section to address your concerns and include a more detailed explanation of the study approach.
The revised section includes a step-by-step breakdown of the study approach, including detailed processes for tissue preparation, staining, imaging, and evaluation procedures. We have also provided additional details on the statistical analysis performed and the staining methods with relevant references. Furthermore, we have provided justifications for the methodological choices made throughout the study to enhance clarity and transparency for readers.
“ 2.1. Tissue selection, staining and imaging
A total of 36 frozen breast tissue samples, comprising 20 non-neoplastic and 16 neoplastic specimens, were acquired from the biobank of our institution once the study had been evaluated by the review board and informed consent had been obtained. The study adhered to the guidelines set by the Ethics Committee.
The neoplastic cases predominantly consisted of invasive ductal carcinomas (n=12) (Figures 1-3), but also included one case each of phyllodes tumour, ductal carcinoma in situ, invasive lobular carcinoma and mucinous carcinoma (Figure 4).
Each sample was completely immersed in saline solution and stained in 50% acetic acid for 20 seconds; this was followed by a wash in saline solution and additional staining in 1 mmol/L acridine orange for 20 seconds. Acridine orange is a dye that binds to nucleic acids and emits green fluorescence when attached to DNA and RNA, with a peak fluorescence emission when excited at 488 nm [26,27]. Acetic acid was used to enhance nuclear detail by inducing the compaction of chromatin, which improves reflectance signals and visualisation. This dye combination increases the visibility of both stromal and tumour structures, enhancing the contrast and definition of the tissue architecture in EVFCM without damaging the samples for subsequent histopathological analysis [28].
For imaging, the samples were scanned using the VivaScope®2500M-G4 (Mavig GmbH, Munich, Germany; Caliber I.D., Rochester, NY, USA) device [29]. This device incorporates dual lasers that operate at distinct wavelengths: fluorescence (488nm) and reflectance (638 nm). Scanning was performed using the fusion mode, which merges the fluorescence and reflectance channels to generate a comprehensive, high-resolution image of the tissue architecture and cellular morphology. This mode is especially useful for distinguishing between neoplastic and non-neoplastic tissue structures, enhancing the information provided by the acetic acid and acridine orange stains.
After scanning, the samples underwent routine histopathological processing, including fixation in formalin, embedment in paraffin, tissue sectioning, and staining with H&E for definitive diagnosis.
2.2. Confocal image evaluation and pathologist assessments
Three evaluators independently assessed the EVFCM images: a Mohs surgeon dermatologist experienced in EVFCM with no prior experience in breast pathology (evaluator A) and two expert breast pathologists with no prior experience in EVFCM (evaluators B and C). The evaluators were blinded to the final histopathological diagnoses, and they classified the images as either neoplastic or non-neoplastic. The evaluators were instructed to rely solely on the EVFCM images for their assessments, and no list of potential histological subtypes was provided; this was to avoid bias in the interpretation.
Each sample was evaluated twice by each pathologist, with a 12-month washout period between the two evaluations to prevent recall bias. Prior to the second evaluation, one of the pathologists (evaluator B) underwent a brief, 4-hour training session on the appearance and morphology of cutaneous carcinomas in EVFCM. In contrast, evaluator C did not receive additional training, but was allowed to examine all the histopathology samples without their corresponding confocal images. The pathologists were asked about the presence or absence of neoplasia during both evaluations and, if observed, the subtype of the visualised neoplasia was noted.
The evaluators were not shown the histopathology results until all evaluations were completed. After the second confocal evaluation, the pathologists reviewed the permanent H&E slides, and the final consensus diagnosis was determined based on the H&E findings. Each EVFCM image was compared to the corresponding histopathological section to confirm the accuracy.
2.3. Statistical analysis
An exploratory data analysis was conducted using frequency and percentage distributions. The sensitivity, specificity, positive predictive value (PPV) and negative predictive value (NPV) were calculated [30]. To assess the correlation and agreement between the evaluators, the Kappa index of concordance was computed [31]. The final evaluation of the H&E-stained slides produced by the two expert pathologists served as the gold standard. A significance level of 5% was adopted. The PASW statistical software (SPSS Corp, Chicago-USA) was used for data analysis. “
- Byvaltsev, V. A.; Bardonova, L. A.; Onaka, N. R.; Polkin, R. A.; Ochkal, S. V.; Shepelev, V. V.; Aliyev, M. A.; Potapov, A. A. Acridine Orange: A Review of Novel Applications for Surgical Cancer Imaging and Therapy. Front. Oncol. 2019, 9, 925. https://doi.org/10.3389/fonc.2019.00925.
- Mathieu, M. C.; Ragazzi, M.; Ferchiou, M.; van Diest, P. J.; Casiraghi, O.; Lakhdar, A. B.; Labaied, N.; Conversano, A.; Abbaci, M. Breast Tissue Imaging Atlas Using Ultra-Fast Confocal Microscopy to Identify Cancer Lesions. Virchows Arch. 2024, Online ahead of print. https://doi.org/10.1007/s00428-024-03783-y.
- Pérez-Anker, J.; Ribero, S.; Yélamos, O.; García-Herrera, A.; Alos, L.; Alejo, B.; Combalia, M.; Moreno-Ramírez, D.; Malvehy, J.; Puig, S. Basal Cell Carcinoma Characterization Using Fusion Ex Vivo Confocal Microscopy: A Promising Change in Conventional Skin Histopathology. Br. J. Dermatol. 2020, 182 (2), 468-476. doi: 10.1111/bjd.18239.
- MAVIG. VivaScope 2500M-G4 Technical Data. Munich, Germany: MAVIG GMBH. Available online: https://vivascope.de/wp-content/uploads/2023/03/VivaScope_Pathology_ExVivo-2500-Datasheet_EN.pdf (accessed on 25 August 2024).
- Altman, D. G.; Bland, J. M. Diagnostic Tests. 1: Sensitivity and Specificity. BMJ 1994, 308 (6943), 1552. doi: 10.1136/bmj.308.6943.1552.
- Landis, J. R.; Koch, G. G. The Measurement of Observer Agreement for Categorical Data. Biometrics 1977, 33 (1), 159–174. PMID: 843571.
We believe these revisions offer a comprehensive explanation of the study methodology and align with your expectations.
- Comment 6. How specificity, sensitivity and Kappa values are computed?
Response 6:
Thank you for raising this question. The statistical analysis section has been revised to include two references related to the calculation of sensitivity, specificity, and kappa, which explain the methodology used for these measures.
- Altman, D. G.; Bland, J. M. Diagnostic Tests. 1: Sensitivity and Specificity. BMJ 1994, 308 (6943), 1552. doi: 10.1136/bmj.308.6943.1552.
- Landis, J. R.; Koch, G. G. The Measurement of Observer Agreement for Categorical Data. Biometrics 1977, 33 (1), 159–174. PMID: 843571.
You can find the revised Statistical Analysis section in the manuscript as follows:
“An exploratory data analysis was conducted using frequency and percentage distributions. The sensitivity, specificity, positive predictive value (PPV) and negative predictive value (NPV) were calculated [30]. To assess the correlation and agreement between the evaluators, the Kappa index of concordance was computed [31]. The final evaluation of the H&E-stained slides produced by the two expert pathologists served as the gold standard. A significance level of 5% was adopted. The PASW statistical software (SPSS Corp, Chicago-USA) was used for data analysis.”
We hope these this adjustment enhances the clarity and understanding of the statistical analysis section. We would like to express our gratitude to you for raising this question, as it allowed us to ensure that the methodology is accurately reflected and properly referenced.
- Comment 7. References 6-12 are too old, the authors should consider to replace the recent references.
Response 7:
Thank you for this valuable suggestion. We have carefully reviewed the references listed in the manuscript and have updated them to newer, more recent studies that preserve the right information we intended to convey. Specifically, we have replaced old references 6-12 as per your suggestion, with recent studies that align the current state of the affected margins. The newly incorporated references now provide a more relevant foundation for the manuscript, particularly in the context of breast-conserving surgery and margin assessment, as demonstrated in the revised text:
“A significant proportion of breast cancer patients undergoing breast-conserving surgery require additional procedures due to positive margins, with at least 20% needing more than one surgery as part of their treatment plan [6-8].”
The updated references supporting this section are as follows:
- Brachtel, E. F.; Johnson, N. B.; Huck, A. E.; Rice-Stitt, T. L.; Vangel, M. G.; Smith, B. L.; Tearney, G. J.; Kang, D. Spectrally Encoded Confocal Microscopy for Diagnosing Breast Cancer in Excision and Margin Specimens. Lab. Invest. 2016, 96 (4), 459–467. https://doi.org/10.1038/labinvest.2015.158.
- Conversano, A.; Abbaci, M.; van Diest, P.; Roulot, A.; Falco, G.; Ferchiou, M.; Coiro, S.; Richir, M.; Genolet, P. M.; Clement, C.; Casiraghi, O.; Lahkdar, A. B.; Labaied, N.; Ragazzi, M.; Mathieu, M. C. Breast Carcinoma Detection in Ex Vivo Fresh Human Breast Surgical Specimens Using a Fast Slide-Free Confocal Microscopy Scanner: HIBISCUSS Project. BJS Open 2023, 7 (3), zrad046. DOI: 10.1093/bjsopen/zrad046.
- Rakha, E. A.; Quinn, C.; Masannat, Y. A.; Lee, A. S.; Tan, P. H.; Karakatsanis, A.; Matrai, Z. T.; Al Shaibani, S. H. M.; Gehani, S. A.; Shaaban, A.; Khout, H.; Chagla, L.; Cserni, G.; Varga, Z.; Yong, W. F.; Meattini, I.; Kulka, J.; Yang, W.; Tse, G. M.; Pinder, S. E.; Fox, S.; Dixon, J. M. Revisiting Surgical Margins for Invasive Breast Cancer Patients Treated with Breast Conservation Therapy—Evidence for Adopting a 1 mm Negative Width. Eur. J. Surg. Oncol. 2024, 50 (10), 108573. https://doi.org/10.1016/j.ejso.2024.108573.
We believe these updates significantly enhance the manuscript’s currency and relevance. Thank you once again for your recommendation.
Sincerely,
Dr. Daniel Humaran Cozar, MD
Principal and Corresponding Author
General and Digestive Surgery Department, Hospital Universitari Germans Trias i Pujol, Badalona (Barcelona, Spain)
Department of Surgery, Universitat Autònoma de Barcelona (UAB)
Phone: (+34) 605 947 014
Email: dhumaranc.germanstrias@gencat.cat

Reviewer 2 Report
Comments and Suggestions for Authors
The paper titled "Unveiling a Surgical Revolution: Single-blind Assessment of Conventional Histology and Ex-vivo Fusion Confocal Microscopy (EVFCM) in Breast Cancer Surgery" explores the diagnostic accuracy of EVFCM in breast cancer surgeries. The authors conducted an observational study with 36 breast tissue samples (16 neoplastic and 20 non-neoplastic) to compare the efficacy of EVFCM with conventional histology. The study used both trained and untrained evaluators to assess EVFCM images, with a focus on their ability to detect neoplasia and differentiate histologic subtypes. The results showed high concordance rates between EVFCM and traditional histology. Sensitivity and specificity metrics were strong, with evaluators achieving over 90% in both areas. The paper suggests that EVFCM could revolutionize breast-conserving surgeries by providing real-time intraoperative margin assessments, reducing the need for reoperations.
The topic is highly relevant and of significant interest to both the surgical and pathology communities, as it addresses key challenges in intraoperative margin assessment. However, there are several points that need to be clarified or expanded upon in a revision to strengthen the paper.
- The methodology regarding how tissue was stained and processed for EVFCM images, especially the use of acetic acid and acridine orange, is not thoroughly explained in terms of how these reagents influence image quality. I recommend to include a brief explanation of the mechanism by which acetic acid and acridine orange improve image contrast and enhance the identification of neoplastic tissues. This will help readers unfamiliar with the technique to better understand the staining protocol.
-While the study evaluates pathologists' ability to recognize neoplasia, it is not clearly specified why evaluator C showed poorer results without additional training, other than confusion. More explanation about the importance of the learning curve for EVFCM could clarify this. I suggest to provide more details on the learning curve associated with EVFCM and how short training can significantly impact performance. You can compare with other studies showing similar trends in learning the technique.
- The paper provides a thorough review of relevant literature on EVFCM but misses opportunities to discuss the potential impact of virtual tools for patient management in breast cancer surgeries. I suggest to incorporate and discuss the study by Żydowicz et al., "Navigating the Metaverse: A New Virtual Tool with Promising Real Benefits for Breast Cancer Patients". This paper, although focusing on virtual technology, complements the discussion on advanced surgical tools like EVFCM by offering a broader perspective on technological innovations in patient care and recovery. You could suggest a section discussing technological advances in breast cancer treatment that covers both EVFCM and virtual tools.
- The conclusion is well-written but could emphasize the broader clinical implications of adopting EVFCM more strongly, especially in cost reduction and patient outcomes. You should highlight how widespread adoption of EVFCM could lead to better healthcare outcomes by reducing re-excisions and healthcare costs, improving cosmetic outcomes, and enhancing the psychological well-being of patients due to fewer surgeries.
Author Response
The paper titled "Unveiling a Surgical Revolution: Single-blind Assessment of Conventional Histology and Ex-vivo Fusion Confocal Microscopy (EVFCM) in Breast Cancer Surgery" explores the diagnostic accuracy of EVFCM in breast cancer surgeries. The authors conducted an observational study with 36 breast tissue samples (16 neoplastic and 20 non-neoplastic) to compare the efficacy of EVFCM with conventional histology. The study used both trained and untrained evaluators to assess EVFCM images, with a focus on their ability to detect neoplasia and differentiate histologic subtypes. The results showed high concordance rates between EVFCM and traditional histology. Sensitivity and specificity metrics were strong, with evaluators achieving over 90% in both areas. The paper suggests that EVFCM could revolutionize breast-conserving surgeries by providing real-time intraoperative margin assessments, reducing the need for reoperations.
The topic is highly relevant and of significant interest to both the surgical and pathology communities, as it addresses key challenges in intraoperative margin assessment. However, there are several points that need to be clarified or expanded upon in a revision to strengthen the paper.
Dear Reviewer,
We would like to express our sincere gratitude for your insightful comments and recommendations, which have greatly contributed to improving the clarity and robustness of our manuscript. To further enhance the quality of the writing throughout the manuscript, we have also submitted it to a professional English editing service.
Below, we provide a detailed response to each of your comments.
- Comment 1. The methodology regarding how tissue was stained and processed for EVFCM images, especially the use of acetic acid and acridine orange, is not thoroughly explained in terms of how these reagents influence image quality. I recommend to include a brief explanation of the mechanism by which acetic acid and acridine orange improve image contrast and enhance the identification of neoplastic tissues. This will help readers unfamiliar with the technique to better understand the staining protocol.
Response 1:
We appreciate your valuable suggestion, as it provides an opportunity to clarify the role of acetic acid and acridine orange in enhancing image quality. In response, we have thoroughly reviewed the entire Material and Methods section to ensure clarity and completeness. We have included the following detailed explanation:
“ Each sample was completely immersed in saline solution and stained in 50% acetic acid for 20 seconds; this was followed by a wash in saline solution and additional staining in 1 mmol/L acridine orange for 20 seconds. Acridine orange is a dye that binds to nucleic acids and emits green fluorescence when attached to DNA and RNA, with a peak fluorescence emission when excited at 488 nm [26,27]. Acetic acid was used to enhance nuclear detail by inducing the compaction of chromatin, which improves reflectance signals and visualisation. This dye combination increases the visibility of both stromal and tumour structures, enhancing the contrast and definition of the tissue architecture in EVFCM without damaging the samples for subsequent histopathological analysis [28].”
To further enrich our response to your valuable comment, we have also included three additional references supporting the mechanisms and effectiveness of the staining protocol.
- Byvaltsev, V. A.; Bardonova, L. A.; Onaka, N. R.; Polkin, R. A.; Ochkal, S. V.; Shepelev, V. V.; Aliyev, M. A.; Potapov, A. A. Acridine Orange: A Review of Novel Applications for Surgical Cancer Imaging and Therapy. Front. Oncol. 2019, 9, 925. https://doi.org/10.3389/fonc.2019.00925.
- Mathieu, M. C.; Ragazzi, M.; Ferchiou, M.; van Diest, P. J.; Casiraghi, O.; Lakhdar, A. B.; Labaied, N.; Conversano, A.; Abbaci, M. Breast Tissue Imaging Atlas Using Ultra-Fast Confocal Microscopy to Identify Cancer Lesions. Virchows Arch. 2024, Online ahead of print. https://doi.org/10.1007/s00428-024-03783-y.
- Pérez-Anker, J.; Ribero, S.; Yélamos, O.; García-Herrera, A.; Alos, L.; Alejo, B.; Combalia, M.; Moreno-Ramírez, D.; Malvehy, J.; Puig, S. Basal Cell Carcinoma Characterization Using Fusion Ex Vivo Confocal Microscopy: A Promising Change in Conventional Skin Histopathology. Br. J. Dermatol. 2020, 182 (2), 468-476. doi: 10.1111/bjd.18239.
This addition also helps to contextualise the imaging process described in the following passage:
“ For imaging, the samples were scanned using the VivaScope®2500M-G4 (Mavig GmbH, Munich, Germany; Caliber I.D., Rochester, NY, USA) device [29]. This device incorporates dual lasers that operate at distinct wavelengths: fluorescence (488nm) and reflectance (638 nm). Scanning was performed using the fusion mode, which merges the fluorescence and reflectance channels to generate a comprehensive, high-resolution image of the tissue architecture and cellular morphology. This mode is especially useful for distinguishing between neoplastic and non-neoplastic tissue structures, enhancing the information provided by the acetic acid and acridine orange stains. “
We trust that this additional detail provides the clarity requested and will help readers better understand the methodology and its relevance to image quality and tissue differentiation.
- Comment 2. While the study evaluates pathologists' ability to recognize neoplasia, it is not clearly specified why evaluator C showed poorer results without additional training, other than confusion. More explanation about the importance of the learning curve for EVFCM could clarify this. I suggest to provide more details on the learning curve associated with EVFCM and how short training can significantly impact performance. You can compare with other studies showing similar trends in learning the technique.
Response 2:
Thank you for this insightful comment. In response, we have revised and expanded the Discussion section, adapting the writing and providing a more comprehensive explanation of the importance of the learning curve in EVFCM. We conducted a thorough review of the referenced articles and have added an additional reference ([34]) that further enriches the content and clarity of this section. We also conducted a more exhaustive analysis to justify evaluator C’s results, focusing specifically on the role of structured learning in achieving proficiency with EVFCM.
To address your helpful comment, we compared our findings with other relevant studies that highlight the benefits of structured training. We have added the following text to the manuscript:
“ In line with the findings of Panarello [32], Bertoni [33] and Shavlokhova et al. [34], our results suggest that the learning curve for interpreting confocal images is relatively short. Prior knowledge of EVFCM can enhance individuals’ ability to accurately identify neoplasia, although distinguishing specific subtypes can remain challenging for those with limited experience of EVFCM.
Bertoni et al. [33] found similar trends when using EVFCM to evaluate prostate biopsies, where the agreement between EVFCM and the histopathological diagnosis improved significantly after a brief training period. Their kappa values increased from 0.68 and 0.79 in the first evaluation to 0.87 in the second, and the ROC curve improved from 0.87 to 0.93; as such, the agreement rates increased from 86% to 95%. Similarly, Shavlokhova et al. [34] reported that there is a short learning curve for confocal image interpretation in oral tissues, with the agreement increasing from 89% in the first evaluation to 97% in the second; this was accompanied by significant improvements in sensitivity and specificity. In 2023, the HIBISCUSS project [7] demonstrated that skill in breast cancer detection could be rapidly acquired, with a pathologist achieving an accuracy of 99.6% following structured training. Surgeons, after a brief training programme, showed a notable improvement, with their sensitivity increasing from 83% to 98% and specificity rising from 84% to 87%. These trends underscore the potential for skill in the interpretation of confocal images to be rapidly acquired, and further highlight the importance of structured training in enhancing diagnostic accuracy.
Conversely, evaluator C, who did not receive training, showed no improvement in the second evaluation. While confusion stemming from exposure to histopathology samples without their corresponding confocal images may have contributed to these inferior results, it is important to recognise that the lack of structured training likely hindered evaluator C’s ability to accurately interpret the confocal images. Additionally, the 12-month washout period before the second evaluation likely disrupted the continuity of the learning curve for evaluator C. This situation illustrates the critical role of structured training programmes in mastering EVFCM. “
- Conversano, A.; Abbaci, M.; van Diest, P.; Roulot, A.; Falco, G.; Ferchiou, M.; Coiro, S.; Richir, M.; Genolet, P. M.; Clement, C.; Casiraghi, O.; Lahkdar, A. B.; Labaied, N.; Ragazzi, M.; Mathieu, M. C. Breast Carcinoma Detection in Ex Vivo Fresh Human Breast Surgical Specimens Using a Fast Slide-Free Confocal Microscopy Scanner: HIBISCUSS Project. BJS Open 2023, 7 (3), zrad046. DOI: 10.1093/bjsopen/zrad046.
33.Bertoni, L.; Puliatti, S.; Reggiani Bonetti, L.; Maiorana, A.; Eissa, A.; Azzoni, P.; Bevilacqua, L.; Spandri, V.; Kaleci, S.; Zoeir, A.; Sighinolfi, M. C.; Micali, S.; Bianchi, G.; Pellacani, G.; Rocco, B.; Montironi, R. Ex vivo fluorescence confocal microscopy: prostatic and periprostatic tissues atlas and evaluation of the learning curve. Virchows Arch. 2020, 476 (4), 511–520. DOI: 10.1007/s00428-019-02738-y.
34. Shavlokhova, V.; Flechtenmacher, C.; Sandhu, S.; Vollmer, M.; Vollmer, A.; Saravi, B.; Engel, M.; Ristow, O.; Hoffmann, J.; Freudlsperger, C. Ex Vivo Fluorescent Confocal Microscopy Images of Oral Mucosa: Tissue Atlas and Evaluation of the Learning Curve. J. Biophotonics 2022, 15 (2), e202100225. doi: 10.1002/jbio.202100225.
Thank you again for your constructive feedback, which has enabled us to provide a more detailed analysis of the learning curve and its impact on EVFCM diagnostic accuracy. We hope this revised text meets the objectives of your comment.
- Comment 3. The paper provides a thorough review of relevant literature on EVFCM but misses opportunities to discuss the potential impact of virtual tools for patient management in breast cancer surgeries. I suggest to incorporate and discuss the study by Żydowicz et al., "Navigating the Metaverse: A New Virtual Tool with Promising Real Benefits for Breast Cancer Patients". This paper, although focusing on virtual technology, complements the discussion on advanced surgical tools like EVFCM by offering a broader perspective on technological innovations in patient care and recovery. You could suggest a section discussing technological advances in breast cancer treatment that covers both EVFCM and virtual tools.
Response 3:
Thank you for this valuable suggestion. We found the study by Żydowicz et al. to be highly relevant, as it opens up current and future possibilities that may greatly enhance the applicability and optimisation of EVFCM. Following a detailed reading of the article, we incorporated a discussion on the role of virtual platforms, highlighting the potential for integrating artificial intelligence (AI) and deep learning systems.
In response to your suggestion, we expanded the discussion section as it follows:
“ EVFCM enhances both surgical precision and diagnostic accuracy, while virtual platforms such as those proposed by Å»ydowicz et al. [35] could complement these advancements by optimising patient care. Integrating artificial intelligence (AI) and deep learning systems could further enhance EVFCM’s capabilities by automating the identification of neoplastic cells, thus improving diagnostic accuracy and reducing the workload of the pathologist. This application of AI, particularly in intraoperative margin assessments, could decrease the rate of positive margins and re-excisions, maximising the benefits of EVFCM in surgery. The integration of AI, EVFCM, and virtual platforms offers a pathway for transforming breast cancer care into a more precise, efficient, and patient-centred model. “
- Żydowicz, W.M.; Skokowski, J.; Marano, L.; Polom, K. Navigating the Metaverse: A New Virtual Tool with Promising Real Benefits for Breast Cancer Patients. J. Clin. Med.2024, 13, 4337. https://doi.org/10.3390/jcm13154337
We hope this revised text meets the objectives of your comment and provides the broader perspective you recommended.
- Comment 4. The conclusion is well-written but could emphasize the broader clinical implications of adopting EVFCM more strongly, especially in cost reduction and patient outcomes. You should highlight how widespread adoption of EVFCM could lead to better healthcare outcomes by reducing re-excisions and healthcare costs, improving cosmetic outcomes, and enhancing the psychological well-being of patients due to fewer surgeries.
Response 4:
Thank you for your valuable feedback. In response, we have expanded the conclusions section to emphasise the broader clinical implications of EVFCM adoption. We have added the following passage to address the potential impact on cost reduction, patient outcomes, and overall quality of care:
“ The widespread adoption of EVFCM could significantly improve healthcare outcomes by reducing re-excisions, which would lower healthcare costs and improve cosmetic results for patients. By minimising the need for additional surgeries, EVFCM could also enhance recovery and psychological well-being of patients, as fewer surgeries would reduce the physical and emotional toll on patients. These combined benefits highlight the potential of EVFCM to transform breast cancer surgery, offering clinical, economic, and psychological advantages that enhance the overall quality of patient care. We believe that EVFCM could play a pivotal role in revolutionising the paradigm of breast cancer surgery. “
We hope this additional detail aligns with the objectives of your comment and provides a stronger emphasis on the clinical, economic, and psychological benefits of EVFCM.
Sincerely,
Dr. Daniel Humaran Cozar, MD
Principal and Corresponding Author
General and Digestive Surgery Department, Hospital Universitari Germans Trias i Pujol, Badalona (Barcelona, Spain)
Department of Surgery, Universitat Autònoma de Barcelona (UAB)
Phone: (+34) 605 947 014
Email: dhumaranc.germanstrias@gencat.cat

Round 2
Reviewer 1 Report
Comments and Suggestions for Authors
I thank the authors for their hard work to improve the paper's standard as per th Journal metric. The paper may be acceptable in the present form.
Reviewer 2 Report
Comments and Suggestions for Authors
The Authors have addressed in an excellent way all the issues raised during the first revision. The paper is notably improved and now it could be suitable for publication.
Comments on the Quality of English LanguageNo detected issues as regard the english grammar